# PERK-Olating Through Cancer: A Brew of Cellular Decisions

**DOI:** 10.3390/biom15020248

**Published:** 2025-02-08

**Authors:** Laurent Mazzolini, Christian Touriol

**Affiliations:** Centre de Recherches en Cancérologie de Toulouse (CRCT), INSERM UMR-1037, CNRS UMR-5071, Université de Toulouse, Avenue Hubert Curien, Oncopole Entrée C, CS 53717, 31037 Toulouse CEDEX 1, France

**Keywords:** PERK, unfolded protein response (UPR), ER stress, cancer, microenvironment, resistance, cell death

## Abstract

The type I protein kinase PERK is an endoplasmic reticulum (ER) transmembrane protein that plays a multifaceted role in cancer development and progression, influencing tumor growth, metastasis, and cellular stress responses. The activation of PERK represents one of the three signaling pathways induced during the unfolded protein response (UPR), which is triggered, in particular, in tumor cells that constitutively experience various intracellular and extracellular stresses that impair protein folding within the ER. PERK activation can lead to both pro-survival and proapoptotic outcomes, depending on the cellular context and the extent of ER stress. It helps the reprogramming of the gene expression in cancer cells, thereby ensuring survival in the face of oncogenic stress, such as replicative stress and DNA damage, and also microenvironmental challenges, including hypoxia, angiogenesis, and metastasis. Consequently, PERK contributes to tumor initiation, transformation, adaptation to the microenvironment, and chemoresistance. However, sustained PERK activation in cells can also impair cell proliferation and promote apoptotic death by various interconnected processes, including mitochondrial dysfunction, translational inhibition, the accumulation of various cellular stresses, and the specific induction of multifunctional proapoptotic factors, such as CHOP. The dual role of PERK in promoting both tumor progression and suppression makes it a complex target for therapeutic interventions. A comprehensive understanding of the intricacies of PERK pathway activation and their impact is essential for the development of effective therapeutic strategies, particularly in diseases like cancer, where the ER stress response is deregulated in most, if not all, of the solid and liquid tumors. This article provides an overview of the knowledge acquired from the study of animal models of cancer and tumor cell lines cultured in vitro on PERK’s intracellular functions and their impact on cancer cells and their microenvironment, thus highlighting potential new therapeutic avenues that could target this protein.

## 1. Introduction

As neoplastic cells proliferate, they encounter numerous stresses to which they must adapt in order to survive. In the context of the whole organism, cells must acquire the capacity to evade detection and destruction by the immune system, as well as to escape the growth inhibitory signals sometimes originating from surrounding cells [1,2]. Furthermore, the harsh growth conditions experienced by tumor cells due to deficits in the nutrient and oxygen supply resulting from insufficient blood flow within the tumor are also a source of stress. Consequently, these constraints, when combined with the rapid proliferation, elevated rate of protein synthesis, and high metabolic demands of the tumor cell, necessitate substantial adaptations in the cells, including the significant reorientation of energy metabolism, altered lipid biosynthesis, and the reprogramming of gene expression, in order to facilitate survival and growth.

The endoplasmic reticulum (ER) is a critical organelle in eukaryotic cells, responsible for the synthesis, folding, and modification of membrane and secreted proteins [3]. Interestingly, many of the stresses, whether of extracellular or intracellular origin, experienced by tumor cells lead to dysfunctions in protein biosynthesis that result in the increased production of misfolded or incompletely folded proteins in the ER [4]. Eukaryotic cells have evolved a highly sensitive and reactive detection system for the accumulation of misfolded proteins in this compartment, which consists of three independent ER membrane-associated sensors: Inositol-Requiring Enzyme 1 (IRE1α), activating transcription factor 6 (ATF6), and protein kinase RNA-like endoplasmic reticulum kinase (PERK). Once activated, these sensors will initiate distinct signaling cascades that collectively work to restore ER function and cellular equilibrium. This process is known as the unfolded protein response (UPR) [5,6,7]. In cancer cells, this protective response is harnessed during cell transformation and tumor development to enable the evasion of various stresses and injuries that impact cell viability and survival. However, when the cell’s protective functions are overwhelmed, the UPR will trigger proapoptotic pathways, allowing damaged cell clearance [8]. It has been shown that the three UPR components, ATF6, IRE1α, and PERK, can contribute to cell death pathways. However, the role of ATF6 appears less direct in this case. In contrast, the available data demonstrate the major function of PERK in this context [9].

In this review, we initiate our discussion with the UPR response and the pivotal function of PERK during UPR activation. We then proceed to provide a comprehensive account of the molecular functions performed by this kinase in cancer cells and their significance for tumor growth. Subsequent to this, we undertake a thorough examination of the outcomes and preliminary insights derived from PERK, targeting studies that have been conducted in tumor models. Finally, we engage in a discussion that encompasses the potential benefits and the technical intricacies associated with PERK inhibition or activation in cancer therapy.

## 2. Endoplasmic Reticulum Stress, the Unfolded Protein Response, and PERK Pathway Activation

It has been estimated that approximately one-third of all proteins are adressed to the ER for the processes of folding and maturation. When the ER’s capacity to fold proteins is overwhelmed, the accumulation of unfolded or misfolded proteins occurs. This stress triggers the unfolded protein response (UPR), which aims to restore homeostasis by enhancing protein folding, reducing protein synthesis, and promoting cell survival pathways [5,6,7]. During cancerous growth, UPR activation helps tumor cells to adapt to their exacerbated proliferation in hostile environmental conditions. However, accumulating data have demonstrated that UPR activation can also switch from a cell protective (i.e., “adaptive” UPR) to a death-promoting program (i.e., “terminal” UPR) when cellular homeostasis cannot be maintained or restored [10]. Consequently, the UPR emerges as a pivotal factor in cancer biology, influencing tumor initiation, progression, and response to therapy [8,11]. The UPR is initiated by three ER membrane-associated proteins: IRE1α, ATF6, and PERK (Figure 1). Interestingly, although these three proteins drive relatively distinct downstream cellular responses, they share a very similar mechanism of activation in response to the accumulation of misfolded proteins in the ER, which is based on their dissociation from the protein BiP/GRP78. BiP (binding immunoglobulin protein), also known as GRP78 (78 kDa glucose-regulated protein) or HSPA5 (heat shock 70 kDa protein 5), is an essential heat shock protein chaperone of the ER that prevents the accumulation of misfolded or unfolded proteins in this cell compartment. In parallel to its protein folding function, BiP/GRP78 also plays an essential role in the triggering of the UPR. Under normal conditions, a fraction of the BiP/GRP78 proteins is indeed bound to the luminal domains of each of the three UPR mediators, PERK, IRE1α, and ATF6. This association maintains these proteins in an inactive state. However, an accumulation of unfolded or misfolded proteins can outcompete the BiP/GRP78 chaperone for binding to these UPR mediators. The subsequent induced release of BiP/GRP78 from PERK, IRE1, and ATF6 allows their activation [5,12]. By regulating all three sensors, BiP/GRP78 ensures the synchronized and fine-tuned activation of the three pathways of the UPR signaling cascade (Figure 1) [5].

### 2.1. The Activation of the ATF6 Pathway

ATF6 is a type II transmembrane protein that plays a significant role in maintaining cellular homeostasis by regulating the transcriptional response to ER stress [13]. Under the conditions of ER stress, ATF6 is activated by a series of molecular events that will enable it to function as a transcription factor. ATF6 activation starts by dissociating from BiP/GRP78, allowing it to migrate from the ER to the Golgi apparatus, where it undergoes two sequential proteolytic cleavages by site-1 protease (S1P) and site-2 protease (S2P) (Figure 1). These cleavages release the N-terminal cytosolic domain of ATF6, known as ATF6f, which contains a basic leucine zipper (bZIP) transcription factor domain [14]. The released active ATF6f form then translocates to the nucleus, where it binds to specific ER stress-response elements in the promoters of its target genes, thus allowing the activation of these genes, which encode, in particular, proteins involved in polypeptide folding, such as chaperones and foldases. Additionally, ATF6f promotes the expression of genes like *EDEM* (ER-degradation-enhancing α-mannosidase-like protein) and *HERP* (homocysteine-induced ER protein), which are involved in the degradation of misfolded proteins in the ER. This process is known as the endoplasmic reticulum-associated protein degradation (ERAD) pathway, and it contributes to the alleviation of ER stress by reducing the unfolded protein load [15,16]. ATF6f also plays a role in maintaining calcium homeostasis by activating the transcription of the calcium pump SERCA (sarcoendoplasmic reticulum calcium ATPase), and it supports ER expansion through the upregulation of the genes involved in lipid biosynthesis; therefore, it increases the capacity of this cell compartment to manage increased protein loads [17]. ATF6 has previously been reported to activate proteins such as the transcription factor CHOP (C/EBP homologous protein), a protein that may play an essential role in apoptotic cell death during terminal UPR [18]. Recent data strongly suggest that its regulatory action on CHOP expression is initially aimed at restricting the expression of this factor under mild stress conditions [19]. Consequently, ATF6 appears to contribute primarily to cytoprotection in the context of cancer, as initially observed [20].

### 2.2. The Activation of the IRE1α/RIDD/XBP1s Pathway

IRE1α is a type I transmembrane protein that contains both a serine/threonine kinase domain and an endoribonuclease (RNase) domain. The standard model of IRE1 activation states that, under normal conditions, IRE1α exists as an inactive monomer. However, during ER stress, the dissociation of the chaperone protein GRP78/BiP from its luminal domain induces a series of conformational changes that allow IRE1α to self-assemble into dimers and oligomers (Figure 1). This facilitates the autophosphorylation of IRE1α, where adjacent IRE1α molecules phosphorylate each other. Although the actual IRE1α activation processes may slightly differ [5,21], autophosphorylation remains the key regulatory step that enhances the endoribonuclease activity of IRE1α, enabling it to engage in two primary actions: first, the induction of Regulated IRE1-Dependent Decay (RIDD), and second, the maturation of the mRNA encoding the X-box binding protein 1 (XBP1) transcription factor through an unconventional cytoplasmic splicing process (Figure 1).

-***RIDD***—The IRE1α-mediated RIDD process dictates the selective degradation of ER-localized mRNAs and microRNAs [22,23]. This initially helps alleviate the protein folding load by reducing the synthesis of new proteins and is crucial for cell survival during transient ER stress. However, under more severe stress conditions, IRE1α can form oligomeric structures showing increased RNase activity. The resulting overactivation of the RIDD process extends the specificity of degradation reactions to other classes of mRNAs, particularly mRNAs encoding antiapoptotic factors, and consequently, RIDD shifts from a protective to a proapoptotic role [24]. Furthermore, the oligomerization of IRE1α can also induce an interaction with the TRAF2 (Tumor Necrosis Factor Receptor-Associated Factor 2) protein on its cytosolic side and the subsequent activation of the c-Jun N-terminal kinase (JNK) pathway (Figure 1) and the induction of proapoptotic signaling [25,26]-***The Activation of XBP1s expression***—Besides its general role in RNA degradation, the IRE1α nuclease can also specifically dictate the cytoplasmic maturation of XBP1 mRNA, allowing its translation into an active transcription factor, XBP1s (Figure 1). This step involves a very unconventional cytoplasmic “splicing” step, in which IRE1α’s RNase domain catalyzes the excision of a 26-nucleotide sequence within the coding region of the unspliced XBP1 (XBP1u) mRNA [27]. The ligation of the cleaved mRNA arms is completed by the tRNA ligase RTCB, resulting in the production of a spliced XBP1 (XBP1s) transcript [28]. This splicing leads to a translational frameshift in the coding sequence of the XBP1s mRNA, allowing the synthesis of an extended, stabilized, and active transcription factor, XBP1s (Figure 1). In parallel to the ATF6f factor, XBP1s upregulates the expression of a number of genes involved in protein folding, ERAD, and lipid biosynthesis. These processes contribute to restore homeostasis, and the pro-survival effects of XBP1s expression in response to ER stress are widely documented [29]. However, the proapoptotic effects associated with XBP1s deregulation have also been described [30]. Indeed, this transcription factor has been shown to upregulate the expression of essential apoptotic factors such as KLF9 (Kruppel-like factor 9) [31]. Therefore, XBP1s can also contribute to the induction of cell death under specific cellular stress conditions.

### 2.3. The Activation of the PERK Pathway

The mode of activation of the ER transmembrane kinase, PERK, appears very similar to that of IRE1α. Under normal conditions, PERK is expressed as an inactive, monomeric form complexed to the BiP/GRP78 chaperone on its luminal domain (Figure 1). However, under the conditions of ER stress, the release of bound BiP/GRP78 will allow the assembly of PERK monomers into higher-order oligomers, a step that facilitates its subsequent autophosphorylation at multiple residues, including threonine 982 (T982), in the activation loop of the kinase domain and tyrosine 619 (Y619), which appear critical [5,32]. These phosphorylations allow the full activation of PERK, which, in turn, enables the efficient phosphorylation of its substrates (Figure 1). This process induces a wide range of downstream cellular responses that have a direct impact on cell viability and survival. The following section will present these different cellular responses in more detail.

## 3. PERK-Activated Cellular Responses and Signaling Pathways

PERK activation triggers a series of intracellular processes that can positively or negatively impact cell survival in response to stress. These various processes, which can be both overlapping and sequential, consist of the following: 1—the attenuation of global translation through the phosphorylation of the α subunit of the eukaryotic translation initiation factor 2; 2—the induction of the selective expression of stress response genes; 3—the activation of a cell-protective process against oxidative stress; 4—the inhibition of cell cycle progression; 5—the maintenance of cell homeostasis through FOXO3 phosphorylation; 6—the lipid-mediated modulation/activation of pro-survival signaling pathways allowing cell protection; 7—the switching of the cell’s response to stress from cytoprotective to cytotoxic when cell injuries cannot be counteracted. All these processes are outlined in the following sections.

### 3.1. PERK Action on Global mRNA Translation

The eukaryotic initiation factor 2 (eIF2) complex is critical for the initiation of translation. It is composed of three subunits, termed eIF2α, eIF2β, and eIF2γ. In association with GTP and initiator methionyl-tRNA (Met-tRNAi), eIF2 forms a structure called the ternary complex which, as part of the ribosomal 43S preinitiation complex, will allow the binding of mRNA through the 5′-cap structure and, after mRNA scanning, the recognition of the mRNA initiation (start) codon. Upon the recognition of the start codon, GTP is hydrolyzed, releasing phosphate and eIF2α-guanosine diphosphate (GDP); this step allows the assembly of the active ribosome through the recruitment of the 60S ribosomal subunit, which is followed by translation elongation and polypeptide synthesis [33,34]. Within the eIF2 complex, GTP hydrolysis is performed by the eiF2α catalytic subunit. Following this step, the eIF2α-GDP complex must then be converted back to eIF2α-GTP in order to regenerate an active 43S complex able to bind to the cap structure and allow the formation of an active ribosome upon the start codon recognition. The GDP to GTP exchange is catalyzed on eIF2α by the guanine nucleotide exchange factor (GEF) eIF2B. However, when the eIF2α subunit is phosphorylated at its serine 51 (Ser51) residue, its affinity for eIF2B is increased, preventing the recycling of eIF2α-GDP to eIF2α-GTP [33,35]. This inhibition disrupts the formation of the ternary complex, leading to the inhibition of cap-dependent mRNA translation initiation. In response to ER stress, the activated PERK rapidly catalyzes the direct and efficient phosphorylation of the eIF2α Ser51 residue (Figure 2A). This effect is pivotal in inducing a global reduction in protein synthesis, thereby helping to alleviate the burden of protein folding within the ER by lowering the influx of new proteins [33,35]. It is important to note that, in addition to the response to ER stress through UPR induction, the inhibition of global mRNA translation constitutes a central protective mechanism used by cells experiencing various additional stresses of either intracellular (e.g., nutrient deprivation or oxidative and mitochondrial stress) or extracellular (e.g., viral infection) origin in order to restore cell homeostasis [36]. In this context, in addition to PERK, three other stress-inducible sensor kinases, able to phosphorylate the Ser51 residue of eIF2α and referred as eIF2α kinases (EIF2AK), have been described: HRI (heme-regulated inhibitor; EIF2AK1), PKR (double-stranded, RNA-dependent protein kinase; EIF2AK2), and GCN2 (general amino acid control non-depressible 2; EIF2AK4) [37]. These kinases and PERK (defined as EIF2AK3), despite their distinct characteristics, exhibit analogous kinase domains and modes of activation, collectively forming the EIF2AK family [38].

**Figure 2 biomolecules-15-00248-f002:**
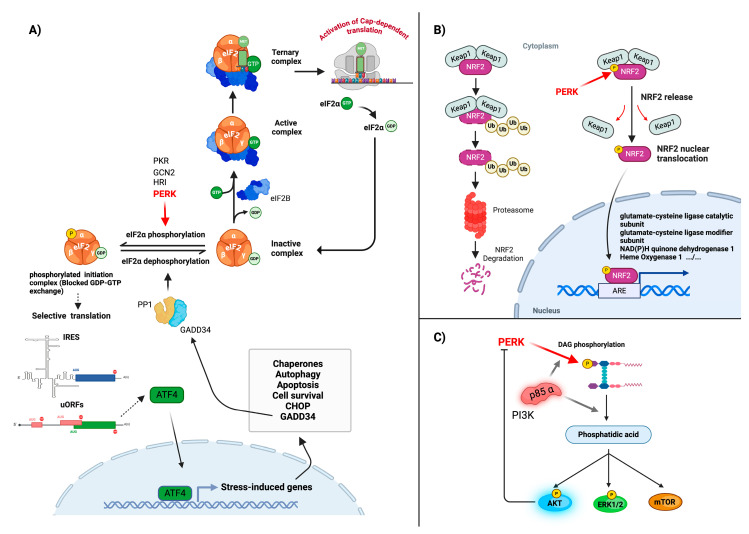
eIF2α, NRF2, and DAG: functions and responses to phosphorylation by PERK. (**A**) The phosphorylation of eIF2α (eukaryotic initiation factor 2α) by PERK plays a critical role in regulating protein synthesis in response to unfolded protein accumulation. This phosphorylation inhibits the conversion of inactive GDP-bound eIF2 into its active, GTP-bound, form by impairing the action of the guanine nucleotide exchange factor eIF2B. Consequently, the formation of the ternary complex and assembly of the 43S ribosomal preinitiation complex is inhibited, resulting in a marked decrease in the overall rate of translation initiation. While inhibiting general cap-dependent translation, eIF2α phosphorylation also selectively promotes the translation of stress-response mRNAs harboring specific regulatory elements, like internal ribosome entry site (IRESs) or upstream open reading frames (uORFs), in their 5′ UTR. One central stress-response mRNA codes for the transcription factor ATF4, which induces the expression of numerous target genes involved in amino acid metabolism, redox homeostasis, and protein folding. Another important ATF4 target gene code for the protein GADD34, which acts as a negative feedback regulator of the stress response by recruiting protein phosphatase 1 (PP1) to eIF2α, thereby promoting its dephosphorylation and the resumption of translational initiation once the stress has been resolved. It is important to note that the phosphorylation of eIF2α can also be performed by three additional eIF2α kinases, PKR, GCN2, or HRI, in response to other stressing conditions. (**B**) Under normal conditions, NRF2 activity is tightly regulated through its interaction with KEAP1, which acts as an adapter for the ubiquitin–ligase complex, leading to NRF2 proteasomal degradation. When activated by ER stress, PERK phosphorylates NRF2, which disrupts its interaction with KEAP1. This leads to the stabilization and nuclear translocation of NRF2, where it can bind to antioxidant response elements (AREs) and activate the transcription of its target genes. This PERK-mediated activation of NRF2 represents an important mechanism by which cells adapt to ER stress and maintain redox homeostasis. (**C**) PERK also possesses an intrinsic lipid kinase activity. PERK can phosphorylate DAG to generate phosphatidic acid (PA), a key lipid-signaling molecule. This process is regulated by the p85α subunit of PI3K (phosphatidylinositol 3-kinase), which enhances PERK’s lipid kinase activity. The production of PA has several downstream effects, including the activation of the AKT, mTOR, and MAP kinase pathways, further regulating cellular proliferation and metabolic adaptation in response to ER stress.

### 3.2. The PERK-Mediated Induction of the Selective Expression of Stress Response Genes

Despite the strong inhibition of global protein synthesis being performed through the canonical cap-dependent pathway, alternative mechanisms will allow the selective translation of mRNAs that code for the proteins that are needed for cell recovery. However, under too harsh or prolonged stress, these mechanisms will also allow the selective translation of mRNAs that code for the proteins that induce cell apoptosis. This selective translation is facilitated by specific cis-acting sequences located in the 5′-untranslated region of their corresponding mRNAs and which are mainly of two types: upstream open reading frames (uORFs) and internal ribosome entry sites (IRESs) (Figure 2A) [39].

***uORFs***—uORFs are very small, open reading frames that precede the initiator codon of the full-length functional protein and interfere with its recognition by the translation preinitiation complex. Within an affected translational context, these uORFs can be bypassed, thus enabling the translation of the downstream coding sequence [40,41]. ATF4 (activating transcription factor 4) plays a crucial role in the UPR. ATF4 functions as a master regulator, controlling the transcription of the key genes essential for directing adaptive functions but also cytotoxic responses when deregulated, during ER stress [42,43]. ATF4 mRNA is a prime example of a transcript that is preferentially translated following PERK activation. The presence of two uORFs in its 5′ untranslated region (UTR) allows for its selective translation when eIF2α is phosphorylated (Figure 2A). Indeed, the second uORF overlaps with the ATF4 coding sequence and acts as a repressive element under normal conditions. However, during stress, the reduced availability of the ternary complex (eIF2-GTP-tRNAmet) causes ribosomes to bypass the second uORF, leading to translation initiation at the ATF4 start codon [40,44]. This results in the production of the transcription factor ATF4, which will primarily activate genes involved in amino acid metabolism, redox homeostasis, and apoptosis.

Another important example of a protein whose translation is induced following stress is the CHOP (C/EBP homologous protein) transcription factor. CHOP is involved in the cellular response to both transient and prolonged ER stress. It plays a critical role in regulating apoptosis by modulating the expression of various proapoptotic factors such as DR5, NOXA, BIM, and PUMA. It also regulates autophagy through the direct modulation of autophagy-related genes (ATGs), either alone or in association with ATF4. CHOP activation can also lead to oxidative stress and cell death if proteostasis cannot be restored. CHOP mRNA is also selectively translated following PERK activation owing to the presence of uORFs in its 5′UTR [40,45]. Moreover, CHOP gene transcription is also directly activated by ATF4, making this protein highly responsive to the UPR activation level.

***IRES***—internal ribosome entry sites are RNA sequences that allow for cap-independent translation initiation through the direct recruitment of the ribosome to the intricate secondary and tertiary structures they form in the 5′UTR. This “internal entry” enables the translation of specific mRNAs, even when global cap-dependent protein synthesis is downregulated [46,47]. It has been demonstrated that several mRNAs that are activated translationally during ER stress possess functional IRES. These proteins have been shown to play critical roles in the regulation of proliferation of cells: their rapid adaptation to stress or, on the contrary, their apoptotic death depends on the cellular and tissue context [39]. Key identified actors include cell cycle and growth regulators, like p27kip [48]; stress-regulatory proteins, like Staufen1 [49], ATF4 (activating transcription factor 4) [50] (Figure 2A), and BiP/GRP78 [51,52]; adaptors to hypoxic stress, such as HIF1α (Hypoxia-Inducible Factor-1α) [53]; antiapoptotic proteins, such as Bcl-2 (B-cell leukemia/lymphoma 2 protein) [54] or XIAP (X-linked inhibitor of apoptosis protein) [55]; as well as proapoptotic factors, such as APAF1 (Apoptotic Protease Activating Factor 1) [47,56].

In conclusion, PERK appears to play a critical role in UPR signaling by initially activating a central, essential, cell protective process set up by the cell to safeguard it against a wide variety of stresses. This process consists of the simultaneous inhibition of overall mRNA translation and the selective expression of genes involved in cell survival. Interestingly, cells have also set up an important downstream regulatory mechanism allowing the dephosphorylation of eIF2α and the subsequent resumption of normal mRNA translation once the stress has been resolved. This process involves the GADD34 protein (growth arrest and DNA damage-inducible protein 34) which, by directly binding to the eIF2α subunit, brings protein phosphatase 1 (PP1) into close proximity with eIF2α, allowing it to dephosphorylate Ser51 (Figure 2A) [57]. It has been demonstrated that the GADD34 gene is directly induced by the transcription factors CHOP and ATF4 (CHOP being itself a direct target gene of ATF4). Strikingly, both CHOP and ATF4 transcription factors show selectively increased translation following the stress-induced inhibition of global protein synthesis (see above). Moreover, the human GADD34 5′UTR contains two non-overlapping uORFs, allowing its active translation during stress. Consequently, the combined upregulation of GADD34 by CHOP and ATF4, in conjunction with the enhanced translation of GADD34 mRNA in response to stress, establishes a robust negative feedback loop that regulates eIF2α phosphorylation levels. This mechanism enables cells to rapidly and precisely adjust their translational activity in response to environmental changes (Figure 2A) [58].

### 3.3. The PERK-Mediated Activation of the Antioxidant Response

Another identified PERK substrate, NRF2 (nuclear factor erythroid 2-related factor 2) [59,60], is a transcription factor that regulates the expression of antioxidant proteins which protect against the oxidative damage triggered by injury and inflammation. Under normal conditions, NRF2 is bound in the cytoplasm to the protein Keap1 (Kelch-like ECH-associated protein 1). Binding to Keap1 sequesters NRF2 in this cell compartment and, in addition, targets it for ubiquitination and subsequent degradation [61]. However, during ER stress, PERK directly phosphorylates NRF2 on threonine 80, inducing its release from Keap1 (Figure 2B). This allows NRF2 translocation into the nucleus, where it will bind to specific DNA sequences known as antioxidant response elements (AREs) located in the promoter regions of genes, allowing their transcriptional activation [60,62]. These NRF2 target genes are involved in antioxidant defense and detoxification and include a glutamate–cysteine ligase catalytic subunit (GCLC), a glutamate–cysteine ligase modifier subunit (GCLM), NAD(P)H quinone dehydrogenase 1 (NQO1), and Heme Oxygenase 1 (HO-1) [63,64,65], as well as superoxide dismutases (SODs) (Figure 2B). By upregulating these antioxidant and detoxifying enzymes, the PERK-NRF2 pathway efficiently helps to neutralize reactive oxygen species (ROS) and other harmful by products of cellular stress [66].

### 3.4. PERK and Cell Cycle Regulation

Blocking cell cycle progression into specific phases is a key mechanism used by cells to halt proliferation following various intrinsic and extrinsic stress. This protective process allows cells to recover from a variety of cell injuries and/or adapt to environmental stressors before continuing with cell division, a highly complex, sensitive process whose successful completion is essential to cell viability. It has been demonstrated for a long time that the induction of the UPR could affect cell cycle progression and that PERK is a critical effector of UPR-induced growth arrest that induces the inhibition of cyclin D1 translation [67]. Since then, increasing amounts of data have highlighted the existing link between ER stress and cell cycle control [68]. Although this regulation involves the three UPR pathways, the PERK branch plays a central role by both regulating the G1 to S (through the downregulation of cyclin D1) and G2 to M transitions (through the selective translation of the P53/47 isoform of P53) [69,70], as well as DNA replication itself (through the claspin-CHK1 activation pathway) [71]. A very recent study has shown that the UPR, mainly acting through PERK, also plays an important role in controlling the undisturbed cell cycle, particularly in ensuring the correct distribution of the endoplasmic reticulum during mitosis [72]. This further highlights the importance of the UPR mediators in the regulation of cell cycle dynamics.

### 3.5. PERK-Mediated FOXO Regulation and Cell Homeostasis

The Forkhead Box O (FOXO) family of transcription factors act as key regulators of cellular homeostasis. The functions of FOXO proteins are diverse and include the regulation of the cell cycle, proteasomal activity, apoptosis, autophagy, DNA repair, and antioxidant response, as well as glucose and lipid metabolism [73,74]. PERK has been found to be able to activate one of its key members, FOXO3 [75], through direct phosphorylation on several amino acid residues [76]. This finding suggests the potential for PERK to function as a cell protector through the process of FOXO protein phosphorylation. In this particular context, PERK’s role appears to be more nuanced. It is recognized that PERK can also indirectly negatively regulate FOXO activity via the activation of AKT signaling. The AKT protein kinase has the capacity to directly inhibit FOXO activity through phosphorylation at several sites [77]. The incidence of PERK on FOXO intracellular function is, therefore, complex and may be contingent on the cellular context, particularly the relative activation levels of the different stress-associated signaling pathways.

### 3.6. PERK and the Phospholipid-Mediated Activation of Pro-Survival Signaling Pathways

PERK also exhibits an intrinsic lipid kinase activity, specifically catalyzing the phosphorylation of diacylglycerol (DAG), resulting in the generation of phosphatidic acid (PA) [78]. The lipid kinase activity of PERK requires the presence of the p85α subunit of phosphoinositide kinase 3, which is itself crucial for the conversion of DAG to PA (Figure 2C). This conversion plays a pivotal role in the activation of several downstream signaling pathways, including mTOR [79,80], Erk1/2 [78,81], and AKT, which are essential for cell survival and proliferation (Figure 2C). Notably, the PERK-mediated activation of AKT establishes a negative feedback loop within the cell [82]: activated AKT, in turn, downregulates PERK activity. This reciprocal regulation establishes an additional sophisticated control mechanism for PERK-mediated action. The unique capability of PERK to function as a lipid kinase reveals the multifaceted role of this kinase beyond traditional protein phosphorylation. In this context, the PERK-directed conversion of DAG to PA may significantly contribute to the regulation of three essential cellular signaling pathways in response to stress.

### 3.7. PERK Overactivation: The Protective-to-Cytotoxic Switch

The data presented above on PERK functions clearly demonstrate that this protein, under moderate, manageable stress conditions act as a barrier against cell death through a variety of cytoprotective molecular mechanisms. Consequently, it is widely admitted that PERK’s action initially favors cancer progression by preventing cancer cell death, as was supported by pioneering studies on the effect of PERK knockout on the in vivo growth of tumors in mice models [83,84]. Nevertheless, the action of PERK is not restricted to cell protection but can also trigger the induction of active cell death processes [7]. This dual effect of PERK, seemingly contradictory in nature, is attributable to the cell’s capacity to have developed additional stress response mechanisms. These mechanisms enable the cell to transition towards a cytotoxic response, aimed at eliminating cells that have sustained irreversible damage, when protective processes become overwhelmed and are unable to restore cellular homeostasis. This “switching” process has been described in the two major cytoprotective pathways induced in the cell: autophagy [85] and the UPR [6,10]. In the case of the UPR, the data currently available clearly show that the increased or prolonged activation of the PERK pathway is a key factor controlling the transition between the protective response (adaptive UPR) and the cytotoxic response (terminal UPR), although the other two pathways are also involved [10].

As for its cytoprotective functions, PERK contributes to increasing the cytotoxicity in the cell by a variety of processes [7,86]. The sustained activation of PERK induces apoptotic cell death processes primarily through the ATF4-mediated upregulation of the CHOP transcription factor [42,87]. Indeed, CHOP plays important roles in the modulation of various pro- and antiapoptotic members of the BCL-2 family protein [87]. Specifically, CHOP has been shown to upregulate the expression of the proapoptotic proteins BIM, PUMA, and NOXA and downregulates the expression of the antiapoptotic proteins BCL-2, BCL-XL, and MCL-1 [88]. The modulation of these different factors altogether induces calcium homeostasis deregulation in the ER, the release of cytochrome C from the mitochondria, and the activation of the caspase 9-mediated apoptotic cascade (Figure 3). The PERK-ATF4-CHOP pathway can also upregulate the expression of the death receptor DR5 (and to a lesser extent DR4) [87,89]. DR5 clustering on the cell surface, in association with adaptor proteins and procaspase-8, will lead to the assembly of the death-inducing signaling complex (DISC) [90]. The formation of DISC then results in the auto-catalytic activation of procaspase-8 into active caspase-8, which, in turn, triggers the execution phase of apoptosis by directly cleaving and activating the downstream effectors caspase-3 and caspase-7 (Figure 3). CHOP has also been reported to induce the transcription of ERO1alpha (ER oxidase 1alpha) [91]. The increased activity of this enzyme in the ER generates a more oxidizing environment in this compartment, triggering calcium release and ultimately increasing the apoptotic response [92].

In addition to the aforementioned role of CHOP, it has been shown that ATF4 can directly contribute to apoptosis induction, specifically by promoting the degradation of the antiapoptotic factor XIAP through the upregulation of ubiquitin–ligases (Figure 3) [93].

Overactive PERK itself can directly impact apoptosis through the modulation of the mitochondrial response. Indeed, PERK has been found to be enriched in specialized regions of the ER which are in close proximity to mitochondria and termed the mitochondria-associated membranes (MAMs). These structures form a physical and functional connection between the ER and mitochondria, which facilitates inter-organelle communication, particularly via the exchange of metabolites such as lipids, reactive oxygen species (ROS), and calcium (Ca^2+^). PERK acts in this compartment by regulating Ca^2+^ exchanges by means of various complementary processes [94]. When PERK is activated, it can, in particular, lead to the release of Ca^2+^ from the ER into the mitochondria. This calcium influx, when excessive, induces a cascade of events in the mitochondria that contributes to triggering the intrinsic apoptotic pathway. These events include the alteration of mitochondrial permeability, the increase of ROS production, the delocalization of proapoptotic proteins to the mitochondria, and the release of cytochrome C into the cytosol. The release of cytochrome C subsequently activates caspase-9, which, in turn, activates effector caspases such as caspase-3, thereby inducing the apoptotic process.

## 4. PERK Kinase and Cancer Cell Progression

Cancer cells have to face many stresses that predominantly originate from the severe growth conditions imposed by their exacerbated proliferation in an often-inappropriate tissue environment. These stresses highly impact their intrinsic viability and ability to proliferate in vivo. By considering the functions performed by PERK with regard to some of the essential processes required for tumor development, altogether defined as the “hallmarks” of cancer [95], it appears that PERK may play major roles in the regulation of several of these hallmarks, which we discuss below (Figure 4). As a first step, an attempt will be made to summarize the different functions that enable PERK to contribute to the intracellular regulation of cancer cell viability. However, it is important to note that the development of a tumor in the body is not solely dependent on the intrinsic survival capacity of the cancer cell. Instead, it is also influenced by its inter-relationships with the immediate cellular environment, its ability to metastasize and invade adjacent tissues, and by the susceptibility of the cancer cells to be recognized and eliminated by the body. In this context, it has been shown that PERK can also play important roles, particularly in the regulation of tumor angiogenesis and in the recognition of cancer cells by the immune system (Figure 4). These additional essential aspects of PERK function will be presented in a second section.

### 4.1. PERK and the Cell Autonomous Modulation of Cancer Cell Survival

Cancer cells are required to cope with a variety of stresses that can threaten their survival and must, therefore, activate protective responses in return. These stresses may be of an extrinsic origin, such as a lack of oxygen (hypoxia) or nutrients [96]. They are also intrinsically associated with the increased metabolic activity of the tumoral cells and their deregulated growth, as in the case of replicative, proteotoxic, or metabolic stress [97,98,99]. PERK plays an essential role in the response (protective or cytotoxic) to these different stresses affecting the cell.

#### 4.1.1. PERK and Hypoxic Stress

Hypoxia, defined as the condition of low oxygen availability, is a pervasive feature of solid tumors. This microenvironmental characteristic has been shown to induce significant physiological stress and to have profound implications for cancer progression and patient outcomes. In addition to limiting the efficacy of therapeutic interventions, hypoxia has been demonstrated to drive the selection of more aggressive cancer cell phenotypes. Extensive clinical research has consistently demonstrated that tumor hypoxic status is a powerful predictor of poor prognosis across various human malignancies [100,101]. Hypoxia has been shown to elicit multiple responses in cancer cells, significantly impacting their survival. A crucial cellular mechanism for sensing and responding to lowered oxygen levels is constituted by the HIF (Hypoxia-Inducible Factor) pathway, which acts through the transcriptional activation of various target genes playing central roles in the adaptation of cells to the low oxygen environment. In addition, hypoxia induces significant physiological stress on the endoplasmic reticulum (ER), resulting in UPR activation. The HIF pathway and the UPR are two intricate signaling pathways that work in concert to finely tune the cellular response of cells to the low oxygen environment [102]. With regard to the UPR signaling pathway, it has been known for a long time that PERK is activated and plays a significant role in tumor cell adaptation and survival [103,104]. Pharmacological inhibitors of PERK have also shown that PERK represents the critical branch of the UPR, allowing hypoxic cells to cope with the stress induced by low oxygen levels [105]. PERK assists in preserving cell homeostasis and viability, primarily by stimulating the translation of ATF4, which acts in multiple parallel ways to achieve this goal [42]. Indeed, ATF4 directs the expression of genes involved in the metabolic switch towards anaerobic energy production, the modulation of amino acid synthesis, the induction of protective autophagy, and the stimulation of protein folding or degradation. Furthermore, ATF4 enhances the expression of genes involved in the modulation of the redox balance and the induction of antioxidant responses in order to mitigate oxidative stress, a common challenge in hypoxic environments [106,107,108]. Recent research has also revealed an unexpected role of PERK in maintaining genomic stability by protecting cells from genomic DNA damage, consisting of the appearance of three-stranded RNA/DNA hybrid (R-loop) structures, which occurs during transcription under hypoxic conditions. This protective effect relies on the PERK/ATF4-mediated induction of the gene coding for senataxin, an RNA/DNA helicase which resolves R-loop structures and allows the avoidance of DNA damage-induced apoptosis [109]. In that study, the pharmacological inhibition of PERK lowered senataxin levels, which resulted in accumulating DNA damage and increased cell death. Collectively, these data clearly indicate that PERK plays an important protective role in the cellular response of cancer cells to hypoxia by triggering a wide range of adaptive mechanisms. Its inhibition could, therefore, represent an attractive therapeutic approach against hypoxic tumors.

#### 4.1.2. The PERK-Mediated Modulation of Mitochondria Energetic Metabolism

It is an established fact that tumoral proliferation is a process that requires a lot of energy. As a result, the production of ATP in cancer cells needs, therefore, to be increased. While the classical mitochondrial oxidative phosphorylation (OXPHOS) process remains the predominant contributor to ATP synthesis in cancer cells, these cells also rely on aerobic glycolysis, despite its comparatively lower energetic efficiency. Cancer cells have generally acquired an improved ability to favor either OXPHOS or glycolysis, depending on the environmental conditions and cellular requirements [110,111]. Moreover, the OXPHOS status of cells was found to either promote or inhibit cell proliferation, depending on the cancer type, although increased OXPHOS activity is generally associated with increased cancer drug resistance [112]. The endoplasmic reticulum and mitochondria have previously been shown to establish direct physical links between each other, as well as intricate bilateral functional relationships [113,114]. Several recent articles have demonstrated that the PERK branch of the UPR pathway plays a central role in this functional dialogue. In response to stress, the protective remodeling of mitochondrial morphology can occur in order to sustain mitochondria integrity and functionality, particularly with respect to ATP production [115]. This process of remodeling, characterized by the elongation of mitochondria and the formation of strongly interconnected networks, has been termed stress-induced mitochondrial hyperfusion (SIMH) [115]. A recent publication [116] demonstrated that PERK was able to induce SIMH through a phospho-eIF2α-dependent process, and that this regulation led to improved OXPHOS-mediated ATP production in the mitochondria and increased cell viability. Interestingly, a previous report implicated the direct PERK target NRF2 in promoting SIMH by favoring the degradation of the mitochondrial fission protein Drp1, although the possible involvement of PERK activation was not investigated in this work [117]. Another study revealed that PERK could increase mitochondrial respiration rates and enhance ATP production by the ATF4-mediated transcriptional activation of the gene coding for SCAF1 (SR-Related, CTD-Associated Factor 1), a factor involved in the assembly of mitochondrial respiratory super-complexes [118]. A recent study [119] demonstrated that in response to glucose deprivation and ER stress, PERK could also covalently associate to the ER oxidoreductase Ero1α. The formation of the PERK-Ero1α complex enhanced the contacts between the endoplasmic reticulum and mitochondria. This facilitated a transfer from the ER to the mitochondria of Ca^2+^, which is essential for activating the key enzymes involved in the Krebs cycle and oxidative phosphorylation. In addition, the increase in ER–mitochondria contact sites was also found to reduce oxidative damage, thereby enhancing cell viability. Collectively, these previously published data unequivocally established PERK as a direct and essential regulator of mitochondrial energetics in response to stress conditions.

### 4.2. The Effects of PERK on the Inter-Relationships Between Cancer Cells and Their Surrounding Environment

The tumor microenvironment (TME) is an intricate and dynamic ecosystem surrounding a tumor, composed of various cellular and non-cellular components that interact with cancer cells [120]. It includes stromal cells (e.g., fibroblasts, immune cells, endothelial cells), blood vessels, the extracellular matrix (ECM), and signaling molecules (e.g., growth factors, cytokines). The TME, through continuous reciprocal interactions with cancer cells, plays a crucial role in tumor growth and spreading. PERK again actively contributes to regulating certain aspects of these interactions, impacting the progression of cancerous cells [121].

#### 4.2.1. PERK Action on EMT, Migration, Invasion, and Metastasis

The epithelial to mesenchymal transition (EMT) is a complex process which allows the cell to acquire numerous new characteristics (altogether defining the “mesenchymal” phenotype) that play important functional roles for cancer metastasis [122]. The EMT facilitates cell detachment from the matrix, while at the same time inhibiting the process of anoikis, a form of programmed cell death induced by the loss of substrate adhesion. Furthermore, the EMT raises the cell’s migratory and invasive potential. It has been reported that under hypoxic conditions, the EMT could be induced by the UPR [123]. This activation is associated with the release of TGF-β (transforming growth factor-beta), a key cytokine involved in the EMT. In this study, the siRNA-mediated knockdown of PERK, ATF4, and ATF6 was found to impair TGF-β production and inhibit the EMT process [123]. Interestingly, the EMT is associated with increased levels of protein synthesis and secretion, accompanied by structural changes in the ER which have been found to activate the PERK branch of the UPR [124]. Consequently, under certain conditions, a cross-regulatory relationship could, therefore, exist between the EMT and the UPR processes. Active PERK/ATF4 signaling appeared to be strongly correlated with EMT-associated gene expression in human patient samples originating from different cancers, including breast cancer, colon cancer, gastric cancer, and lung cancer [124]. Furthermore, PERK/ATF4 signaling has been demonstrated to be essential for efficient invasion and metastasis [124]. The currently available data indicate that the PERK/ATF4-mediated stimulation of metastasis can act both by protecting cells against detachment-induced anoikis and by directly inducing the expression of the protein effectors of cell migration and metastasis. ATF4-mediated protection against anoikis was shown by several studies to involve autophagy and/or stimulation of the antioxidant response [125,126,127]. In an oesophageal squamous cell carcinoma (ESCC) model, the increase in invasiveness was reported to be achieved, at least in part, through the ATF4-mediated induction of the genes coding for the matrix metalloproteinases (MMPs) MMP-2 and MMP-7 [128]. ATF4 was also found to induce the expression of the gene coding for LAMP3 (lysosome-associated membrane protein 3) under hypoxic conditions. LAMP3 is a protein that is frequently overexpressed in cancers and plays an important role in cell migration [129]. In two other reported studies, using, respectively, breast and cervical cancer model systems, ATF4 or PERK inhibition resulted in a significant decrease in cell migration and invasion [130,131] that was associated with LAMP3 downregulation. Furthermore, the involvement of ATF4 in *LAMP3* gene regulation though directly binding to its promoter has recently been demonstrated [132]. In conclusion, the data presented above collectively indicate that PERK/ATF4 signaling can promote metastatic cancer spread by both increasing cell survival and directly upregulating the cellular enzymes involved in the cell migratory and invasive processes.

#### 4.2.2. PERK Action on Tumoral Angiogenesis

Angiogenesis, the formation of new blood vessels from pre-existing ones, is a crucial process in various physiological and pathological conditions. While its role is vital for development and wound healing, it also plays a crucial role in cancer progression, where it sustains tumor growth and metastasis. The involvement of the PERK pathway of the UPR in angiogenesis is particularly noteworthy since it has been previously shown that the translation of several pro-angiogenic factors can specifically be stimulated through IRES-dependent mechanisms following PERK activation and the global inhibition of cap-dependent protein synthesis. This phenomenon is particularly evident in the case of the two major pro-angiogenic factors, VEGF-A (vascular endothelial growth factor A) [133] and FGF-2 (Fibroblast Growth Factor 2) [134,135], as well as of DLL4 (Delta-like ligand 4) [35], another key regulator of angiogenesis. DLL4 is prominently expressed in specialized endothelial cells located at the leading edge of structures called vascular sprouts, which play a critical role in blood vessel formation and remodeling. The translation of these major angiogenic players has been demonstrated to occur both in vitro and in vivo in transgenic mouse models following the activation of the PERK pathway [35,136,137,138]. It is noteworthy that under stress conditions, certain IRES trans-acting factors (ITAFs) have been observed to undergo a translocation from the nucleus to the cytoplasm. This relocation of ITAFs has been shown to depend on the activation of the PERK pathway and has been hypothesized to be partly involved in enhancing IRES-mediated translation [35,47].

In terms of functionality, several lines of evidence suggest that the PERK-mediated activation of the expression of pro-angiogenic factors may significantly contribute to sustained angiogenesis in vivo. Indeed, tumors derived from PERK-proficient (*PERK*+/+) mouse embryonic fibroblasts (MEFs) demonstrate robust micro-vessel formation in mice, characterized by the proliferation and organization of endothelial cells into new blood vessels. Conversely, tumors derived from PERK-deficient (*PERK*−/−) MEFs exhibit a markedly reduced vascular network [139]. Furthermore, studies in mice insulinoma models showed that tumors induced in PERK-knockout individuals exhibit significantly reduced vascularity compared to their wild-type counterparts, even when comparing size-matched tumors [140]. Moreover, in mouse xenograft models of human cancers, such as pancreatic cancer and multiple myeloma, the pharmacological inhibition of PERK by compound GSK2656157 led to decreased vascular density, altered amino acid metabolism, and reduced tumor growth [141]. In a similar manner, the use of mice xenografts of human renal cell carcinoma revealed that pharmacological PERK inhibition by compound HC-5404 enhanced the antiangiogenic effects of vascular endothelial growth factor (VEGF) receptor (VEGFR) inhibitors, resulting in further decreased blood vessel density and improved tumor regression [142].

In conclusion, PERK also emerges as an important regulator of tumor angiogenesis. Through its activation in response to hypoxia and nutrient stress, PERK orchestrates a complex network of cellular responses that ultimately promote blood vessel development within the tumor microenvironment. The observed reduction in tumor vascularity upon PERK deficiency or pharmacological inhibition across various cancer models further highlights this kinase as an attractive target for antiangiogenic therapy.

#### 4.2.3. The Impact of PERK on the Antitumor Immune Response

Endoplasmic reticulum stress and the unfolded protein response have been found to have the capacity to elicit various protective barriers against antitumoral immunity that involve the IRE1-XBP1 and PERK branches of the UPR [143]. In this context, PERK plays multiple roles in the recognition of cancer cells by the immune system. Within the tumor cell itself, PERK can influence the production of molecules that promote its recognition and elimination by the surrounding immune cells (immunogenic cell death). PERK can also regulate the activity and, in some cases, the production of different classes of immune cells that perform essential functions. These include the direct recognition of cancer cells (dendritic cells, macrophages, T cells) and the modulation of the induced immune response (myeloid-derived suppressor cells) [143].

-***PERK and immunogenic cell death***—immunogenic cell death (ICD) is a process characterized by the improved recognition of cancer cell antigens by the organism, which can elicit a robust adaptive antitumor immune response [144]. Besides the requirement of *bona fide* cancer-specific immunogenic epitopes (tumor-associated antigens), ICD triggering also depends on the release by the stressed or dying tumor cell of molecules that will act as adjuvant signals. These molecules stimulate the initiation of the antitumor immune response, particularly by amplifying the production and activity of antigen-presenting cells, such as dendritic cells. The aforementioned adjuvant molecules were collectively referred to as damage-associated molecular patterns (DAMPs). Paraptosis is an alternative form of programmed cell death [145] characterized by cytoplasmic vacuolization and the swelling of the ER and mitochondria. It has been previously suggested that paraptosis could contribute to the production of several protein constituents of DAMPs, as well as others immunostimulatory molecules. A recent study by Mandula et al. [146] highlighted the direct role of PERK in the regulation of the release of ICD drivers in melanoma cells by preventing the induction of paraptosis. In this model, PERK ablation led to paraptosis induction and the ensuing stimulation of ICD.-***The impact of PERK on dendritic cells function***—a central result of the work of Mandula et al. [146] was, however, to demonstrate that *PERK*-null tumors showed increased type I interferon production. This promoted the differentiation of monocyte precursors into a specific subclass of dendritic cells (DCs), the monocyte-derived dendritic cells (moDCs), which are highly efficient in antigen presentation. The increased production of moDC stimulated the immune response in various ways and actively contributed to inhibiting tumor growth in vivo.-***PERK action on tumor associated macrophages***—tumor-associated macrophages (TAMs) represent a further critical component in the immune response against tumors. Despite the heterogeneity of TAM functions within the TME, they have been basically distinguished phenotypically into two classes, considering their impact on tumor growth. The “M1-like” macrophages are generally associated with antitumor immunity, characterized by the production of pro-inflammatory cytokines that can inhibit tumor growth and promote tumor cell death. Conversely, M2-like macrophages have been observed to support tumor growth by enhancing angiogenesis, suppressing adaptive immunity, and facilitating tissue repair and remodeling, thereby favoring tumor invasiveness and metastasis [147,148]. It has been recently demonstrated that PERK could promote the immunosuppressive M2 macrophage phenotype through the enhancement of lipid metabolism and mitochondrial respiration in these cells [149]. This metabolic reprogramming was allowed by the ATF4-mediated direct transcriptional activation of the gene coding for PSAT1 (phosphoserine aminotransferase 1), a key enzyme involved in serine biosynthesis that is essential to mitochondrial and lipid metabolism. Moreover, increased serine biosynthesis also contributed to inducing epigenetic changes involving the demethylase JMJD3 (Jumonji domain-containing protein-3), thereby enabling the activation of the genes required for the development of the M2 phenotype. The pharmacological inhibition of PERK with the drug GSK2656157 reprogrammed TAMs towards the M1 phenotype and enhanced the T cell response against tumor cells in a mouse melanoma model. The study clearly demonstrated that PERK could play a significant role in shaping the behavior and function of tumor-associated macrophages and, consequently, the immune modulation of tumor growth.-***The impact of PERK on T cell function***—T cells, along with B cells, represent the essential effectors of adaptive immunity and play a critical role in the elimination of tumor cells [150]. Two published studies have reported that PERK activation can significantly modulate T cell function by affecting CD8+ cytotoxic T cells, as well as CD4+-regulatory T cells (Treg). Indeed, in the first study, the induction of CHOP expression by the PERK-ATF4 pathway was shown to be directly involved in inhibiting CD8+ cytotoxic T cell function as well as in favoring the induction of apoptosis in these cells. The investigation revealed that CHOP exerts its function by directly repressing the *TBX21* gene encoding T-bet, a central regulator of CD8+ T cell function. The pharmacological inhibition of PERK reduced the CHOP levels and counteracted its effect on T-bet expression, thereby restoring the effector function of the CD8+ T cell [151]. The second report indicated that PERK activation in the TME also enhanced Treg suppressive functions through the ATF4-mediated upregulation of the gene coding for foxp3 (forkhead box P3), a central Treg transcription factor, as well as of genes involved in the production of anti-inflammatory cytokines, like TGF-β and IL-10 [152]. In this study, PERK pharmacological inhibition restored the functionality of effector T cells and enhanced the antitumor immune responses.-***The impact of PERK on MDSCs***—myeloid-derived suppressor cells (MDSCs) are a heterogeneous population of cells that arise during cancer and other pathological conditions. Derived from hematopoietic stem and progenitor cells (HSPCs), these cells are known for their ability to suppress T cell responses and promote tumor growth [153]. Several studies have revealed that PERK can directly increase the activity of MDSCs. In particular, a study by Mohamed et al. demonstrated that PERK enhanced the functionality of MDSCs by activating NRF2 [154]. NRF2 principally acted in MDSCs by upregulating the genes involved in redox homeostasis and mitochondrial function, thereby increasing the mitochondria activity and antioxidant capacities in these cells which both support their immunosuppressive function. The inhibition of PERK has been shown to disrupt NRF2-driven antioxidant capacity and mitochondrial respiratory homeostasis. This, in turn, resulted in the release of mitochondrial DNA in the cytosol which triggered the STING (stimulator of interferon genes)-dependent expression of type I interferons and, consequently, the activation of CD8+ T cells. Therefore, PERK inhibition can switch MDSCs from an immunosuppressive phenotype into a state that is immunostimulatory, thereby augmenting antitumor immunity. Another recent study [155] showed that PERK activation plays a crucial additional role in the production of MDSCs themselves by reprogramming hematopoietic stem/progenitor cells (HSPCs). Liu et al. showed that this process involves the activation of the PERK–ATF4–C/EBPβ signaling pathway. Specifically, C/EBPβ acted by upregulating genes promoting myeloid lineage commitment and MDSC production, as well as by reinforcing MDSC function itself. The inhibition of PERK signaling in the spleen prevented the formation of MDSCs and increased the antitumoral immune response in the studied mouse models.

As highlighted by the preceding data, the induction of PERK has a significant impact on the evasion of antitumor immune responses by cancer cells by the modulation of various immune effector cells. PERK, therefore, appears to function as a central mediator of the defensive barrier raised by tumoral cells against immune system defenses in response to stress. This makes PERK an attractive target for cancer immunotherapy.

## 5. PERK as a Target for Cancer Therapy: Current Status and Possible Future Directions

The following sections will delve deeper into the specific implications of PERK in cancer therapy, attempting to provide a comprehensive understanding of how targeting PERK-mediated pathways could offer new therapeutic opportunities in oncology. The two facets of PERK’s action in the cell: protective or cytotoxic, suggest the use of two possible opposing levers: the inhibition or, alternatively, the activation of this kinase. These two somewhat antagonistic approaches are discussed below, with some thought given to the appropriateness of using one or the other, depending on the type of cancer targeted.

A comprehensive overview of key research studies demonstrating the antitumor potential of PERK targeting is also summarized in Table 1. This table catalogues various human malignancies investigated, the experimental methodologies employed for PERK targeting, the specific drugs utilized, and the observed effects on cancer cells in both in in vitro and in vivo contexts.

### 5.1. Inhibition of PERK in Cancer Therapy

Pioneering work has described the characterization of two pharmacological inhibitors directed against PERK, GSK2606414 [159] and GSK2656157 [141,166]. These studies reported an antitumoral effect of these compounds in human xenograft models of pancreatic adenocarcinoma and multiple myeloma in mice. These observations suggested that the pharmacological targeting of the protective adaptative UPR through the inhibition of the PERK pathway could represent an interesting strategy to control tumor growth. Furthermore, a recent study demonstrated the efficacy of a third PERK inhibitor, HC-5404 (also called HC4 and LY-4), in eradicating dormant disseminated cancer cells (DCCs) that often survive after chemotherapeutic treatment and contribute to metastasis [162]. These initial studies demonstrated that the pharmacological inhibition of PERK had an impact on tumor development in vivo not only by abolishing cancer cell protection against proteotoxic stress but also in certain instances by additional mechanisms, such as the inhibition of angiogenesis or the reactivation of the antitumor immune response. However, to date, the number of studies reporting the in vivo antitumor effects of PERK inhibitors and proposing their use in a single therapy approach remains very limited. However, other reports have described the promising use of these compounds in combination therapy with other drugs. Indeed, in multiple renal cell carcinoma (RCC) tumor models, the PERK inhibitor HC-5404 was reported to significantly enhance the antiangiogenic effect of standard VEGF receptor tyrosine kinase inhibitors (VEGFR-TKIs), such as axitinib and lenvatinib, inhibiting the formation of both new and mature tumor blood vessels [142]. Other promising results were obtained in multiple myeloma (MM) cancer cells. MM cells exhibit notably elevated protein synthesis rates and increased susceptibility to proteotoxic stress. These cells were previously reported to demonstrate heightened sensitivity in vitro to treatment with PERK inhibitors, which induced autophagic cell death [156]. Interestingly, recent studies on this cancer model confirmed that treatment with the PERK inhibitor GSK2606414 alone significantly reduced MM cell proliferation in vitro, as well as in vivo, in mouse xenografts models. Furthermore, the combination of GSK2606414 with the proteasome inhibitor drug bortezomib, an established anti-myeloma compound, resulted in markedly enhanced therapeutic effects. This synergistic approach led to more potent antitumor activity compared to either agent alone [157]. In this combined targeting strategy, PERK inhibition resulted in persistent protein synthesis leading to increased proteotoxic stress. Despite the promising results outlined above, the pharmacological targeting of PERK remains challenging due to two primary issues. Firstly, the inhibition of PERK activity may result in adverse effects in the body by targeting cells that depend on active PERK signaling for proper cell function. This potential limitation is well illustrated by the identification of PERK dysfunction as a causative factor for Wolcott–Rallison syndrome (WRS) [167]. In this disease, the impaired function of PERK affects particularly cells and tissues with high secretory functions, such as the pancreas, liver, and skeletal system, and leads to multiple defects in the body. The potential consequences of PERK inhibition have been particularly documented in the context of pancreatic β-cells, where the loss of PERK or its pharmacological targeting has been shown to result in the impaired folding and processing of proinsulin, the accumulation of misfolded proteins in the ER, and enhanced cell death (for a recent article: [168]). The second limitation of PERK inhibition therapy is that currently available inhibitor molecules may have inappropriate intracellular effects which could partially compromise their therapeutic use. A recent study indeed revealed that the commonly used PERK inhibitors, which function by decreasing PERK’s kinase activity though competitive association to the ATP-binding site of PERK, could, at high concentrations, unexpectedly activate another eiF2α kinase, GCN2/EIF2AK4. This activation triggers an integrated stress response in the cell which could promote cell survival [169]. This unanticipated consequence may present a significant challenge to PERK-targeted therapeutic interventions, wherein the inhibitors typically necessitate administration at comparatively elevated concentrations within a physiological environment. The lack of specificity of these inhibitors may necessitate a more cautious interpretation of their actions, particularly with respect to the precise contribution of PERK inhibition. Indeed, in a mouse model, the PERK inhibitors GSK2606414 and GSK2656157 were found to inhibit the protein RIP1K (receptor-interacting serine/threonine–protein kinase), which is a major intracellular mediator of pro-death and inflammatory stimuli. Treatments with GSK2606414 and GSK2656157 blocked TNF-induced RIPK1 kinase-dependent cell death in vivo, fully restoring mouse survival via the direct inhibition of RIPK1 itself, at concentrations that did not impair PERK signaling [170]. Despite these potential restrictions and drawbacks, the PERK inhibitor HC-5404-FU (an orally bioavailable hemifumarate salt form of HC-5404) is currently undergoing clinical trials, with phase I studies underway to evaluate its safety and efficacy in solid tumors, bringing the PERK inhibition approach closer to potential clinical application (ClinicalTrials.gov ID: NCT04834778).

### 5.2. Promoting the Dark Side of PERK Business: PERK Activation and Cancer Cell Death

While PERK activation is frequently linked to cancer cell survival, many studies have also demonstrated its essential role in inducing cell death when cells’ homeostatic capacities and protective pathways are overwhelmed. Such circumstances may particularly arise in cancers following chemotherapeutic treatments, which can induce cytotoxic UPR signaling. Since PERK is known to play a crucial function in apoptotic cell death induction during terminal UPR [10], its activation could be necessary for the death-inducing effect of some anticancer drugs. Consequently, boosting PERK activity could represent a promising means to improve the therapeutic efficacy of these drugs in various cancer types. For instance, in colorectal cancer, the acute activation of PERK is required for cell death and is mediated by agents such as histone deacetylase inhibitors [171], the anti-inflammatory drugs sulindac [172], or ciclopirox, a wide-spectrum anticancer drug [173]. Again, in colorectal cancer, the pharmacological PERK activator CCT020312 was found to strongly enhance the cytotoxic action of treatment with the antimitotic agent paclitaxel (taxol), both in vitro and in vivo [164]. This finding was reinforced by a very recent study performed in breast cancer cells that demonstrated that taxanes can activate PERK independently of their action on microtubules, and that the activation of the PERK/eIF2α axis is a crucial event for taxane-induced apoptosis. Indeed, the PERK/eIF2α pathway was found to be attenuated in breast cancer cells resistant to paclitaxel (PTX) when compared to paclitaxel-sensitive cells. Remarkably, the reactivation of the PERK/eIF2α axis in these resistant cells using CCT020312 again restored sensitivity to the PTX treatment, both in vitro and in vivo [160]. The unexpected implication of PERK activation for the action of taxanes and the demonstrated synergistic effect of PERK activators and paclitaxel point to a new strategy for improving taxanes’ antitumoral action. This strategy should be applicable to various cancer types since the induction of the PERK pathway by treatment with taxanes was also observed in various other cancer cell lines in the last study. The involvement of PERK in mediating the cytotoxic response to anticancer drugs has been documented in both solid tumors and hematological malignancies, such as leukemia in particular, for which there are many examples in the literature [174]. In particular, in acute myeloid leukemia (AML), several drugs, like JA3 and JA7 [175], VAS3947 [176], CXL146 [177], Camalexin [178], RS-F3 [179], Quizartinib (AC220) [180], Cryptotanshinone [181], Oxalicumone A [182], Miltirone [183], Gossypol (BH3 mimetic) [184], curcumin [185,186], PYZD-4409 [187], Eicosapentaenoic acid [188], and Fenretinide [189], have been demonstrated to induce a cytotoxic response in the cell through terminal UPR induction mediated by PERK activation.

The number of available PERK activators remains limited, as most research studies were initially focused on blocking PERK’s protective functions in tumors. However, the published data indicate that these molecules may possess therapeutic relevance in sensitizing cancer cells to apoptosis by overloading stress pathways and promoting cell death under severe stress conditions, which can be encountered particularly during the response to pharmacological treatments. To the best of our knowledge, no clinical trial has been conducted to date that uses a PERK activator for the purpose of combating tumors, either in isolation or in combination with chemotherapy.

As for the PERK-inhibiting approach, the PERK activation strategy may also have unwanted side effects. Given PERK’s role in cellular responses to stress, systemic PERK activation could adversely affect non-cancerous cells or tissues, particularly by disrupting normal mRNA translation. This could strongly impact the function of cells that are highly dependent on active protein synthesis, such as secretory cells, immune cells, and neurons. In the context of cancerous cells, PERK activation, whether alone or in combination with a pharmacological drug, may prove counterproductive if it does not reach the threshold level required for the induction of the cytotoxic cell response. In such circumstances, PERK’s protective functions may prevail. With respect to the pharmacological treatment itself, a report published in 2017 indicated that increased PERK activity may also, in some situations, be detrimental to the action of the anticancer drug. Indeed, the authors reported that increasing the PERK/NRF2 signaling pathway was the origin of a developed cross-resistance to chemotherapy in human cell lines originating from breast cancer, colon cancer, and osteosarcoma. Indeed, PERK/NRF2 signaling was shown to directly upregulate the expression of MRP1 (multidrug resistance protein 1), a protein of the ABC family of transporter proteins which is capable of actively effluxing a broad range of anticancer drugs from cells [190]. Inadequate PERK activation could, therefore, in some situations, reduce the drug uptake in the cell, thereby severely restricting the drug’s efficacy. Interestingly, in the study, PERK knockdown restored intracellular drug accumulation and chemosensitivity, both in vitro and in vivo. Favoring a PERK inhibition approach (as proposed in the preceding chapter) would, therefore, be a more suitable therapeutic strategy in this context.

## 6. Conclusions

Currently, the available laboratory data clearly indicate that by taking advantage of PERK’s dual role—initially adaptive but cytotoxic in the event of prolonged stress—targeting this kinase makes it possible to achieve better cytotoxic effects in vitro on cancer cells in culture and to induce a more effective antitumor response in vivo in various animal models of cancer. These studies have revealed that PERK targeting can impair tumoral growth by a wide variety of mechanisms, involving the tumor cell itself but also its microenvironment. These include increased tumor cell death through apoptosis or autophagy, decreased metastasis, impaired angiogenesis, reduced resistance against hypoxia, reduced tumor dormancy, enhanced immunogenic cell death, and a restored antitumor immune response. Moreover, since ER stress response and the UPR are activated in a very large number of cancer types, PERK-targeting therapy may be applicable to a wide range of malignancies [8,191]. It may prove to be particularly efficient and appropriate in cancer cells that experience high levels of intrinsic and extrinsic stresses and that may exhibit significant dependence on ER stress and UPR pathways, such as glioblastoma [192] or hematological malignancies [193,194], for example.

Notwithstanding its potential, the therapeutic targeting of PERK requires crucial additional work and studies in order to be validated as a new, effective anticancer strategy. In this context, it appears essential to establish clear milestones in research and development. First, developing selective and potent PERK pharmacological inhibitors/activators that target cancer cells while minimally impacting normal cells and the tissues of the organism remains a significant challenge. Second, conducting long-term safety studies to assess the risks associated with PERK pharmacological modulation will be critical in the development, or not, of this strategy. The parallel deepening of our understanding of the factors that regulate the delicate balance between PERK’s pro-survival and proapoptotic roles in tumoral, as well as normal, cells will also be of great importance for translating laboratory findings into potential new, effective clinical treatments.

## Figures and Tables

**Figure 1 biomolecules-15-00248-f001:**
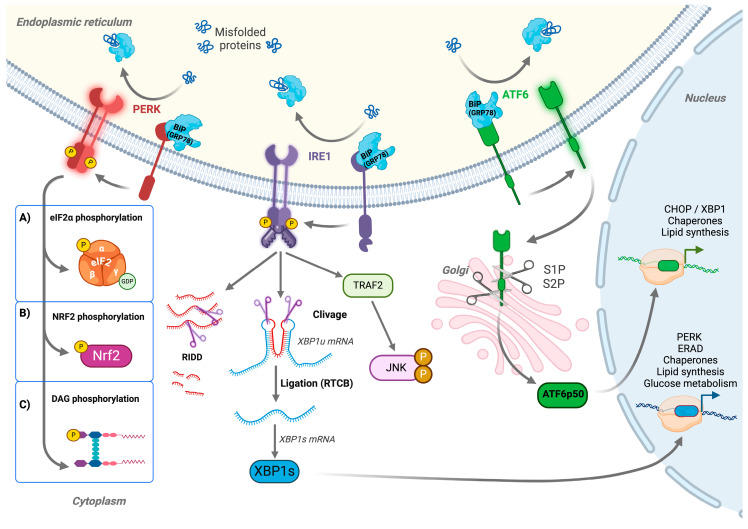
The UPR signaling pathway. The unfolded protein response (UPR) involves three main branches, each initiated by a distinct ER transmembrane sensor: PERK, IRE1, and ATF6. Upon ER stress, these sensors dissociate from the ER-resident chaperone BiP/GRP78, leading to their activation. ATF6, once released from BiP, translocates to the Golgi apparatus where it is cleaved twice, releasing its cytosolic domain that acts as a transcription factor to induce the expression of ER chaperones and ERAD components. IRE1 is a bifunctional protein featuring both a kinase and an RNase domain on its cytosolic side. Once activated, IRE1’s RNase domain performs two critical functions: (i) First, it catalyzes the unconventional splicing of the XBP1 transcript into a matured mRNA form, which is then translated in the active isoform of XBP1: XBP1s. The XBP1s protein then translocates to the nucleus, where it upregulates various UPR target genes, including those encoding ER chaperones and components of the ER-associated protein degradation (ERAD) machinery. (ii) Second, IRE1s RNase domain also mediates the selective degradation of RNAs localized to the ER membrane, resulting in a reduction of protein import into the ER lumen, a process called RIDD (Regulated IRE1-Dependent Decay). IRE1 also recruits TRAF2, forming a complex that subsequently activates the kinase ASK1, which enhances the JNK pathway, a key element in stress-induced apoptosis. The third UPR mediator, *PERK*, once activated, phosphorylates three main substrates, each playing a crucial role in maintaining cellular homeostasis: the proteins eIF2α (A), NRF2 (B) and the lipid molecule diacylglycerol (DAG; C). The precise role of each of these PERK substrates is outlined in Figure 2.

**Figure 3 biomolecules-15-00248-f003:**
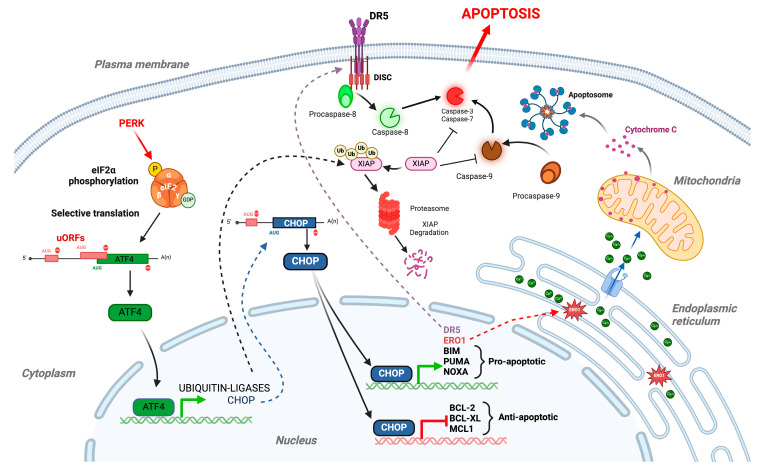
PERK-mediated activation of cell death processes. PERK activation leads to the phosphorylation of eIF2α, which selectively promotes the translation of ATF4. Prolonged ATF4 production can induce the expression of the proapoptotic transcription factor CHOP (C/EBP homologous protein), contributing to apoptotic induction by activating the expression of DR5 (death receptor 5), ERO1α (ER oxidase 1alpha), and the BH3-only proapoptotic proteins BIM, PUMA, and NOXA while inhibiting the expression of the antiapoptotic proteins BCL-2, BCL-XL, and MCL1. DR5 synthesis and activation leads to the maturation of pro-caspase 8 into active caspase-8, which initiates a caspase cascade, leading to the activation of executioners caspase-3 and caspase-7. ERO1α induces, in particular, the release of calcium ions from the ER and their import into the mitochondria, contributing to the mitochondria-mediated induction of apoptosis though the release of cytochrome C in the cytosol, allowing apoptosome assembly and the downstream activation of caspases. The antiapoptotic protein XIAP (X-linked inhibitor of apoptosis protein) protects cells against death by inhibiting both the initiator caspase-9 and the executioners caspase-3 and caspase-7. ATF4 also favors cell death by inducing the expression of ubiquitin–ligases that target XIAP for proteasome-mediated degradation, thereby restoring caspase activation.

**Figure 4 biomolecules-15-00248-f004:**
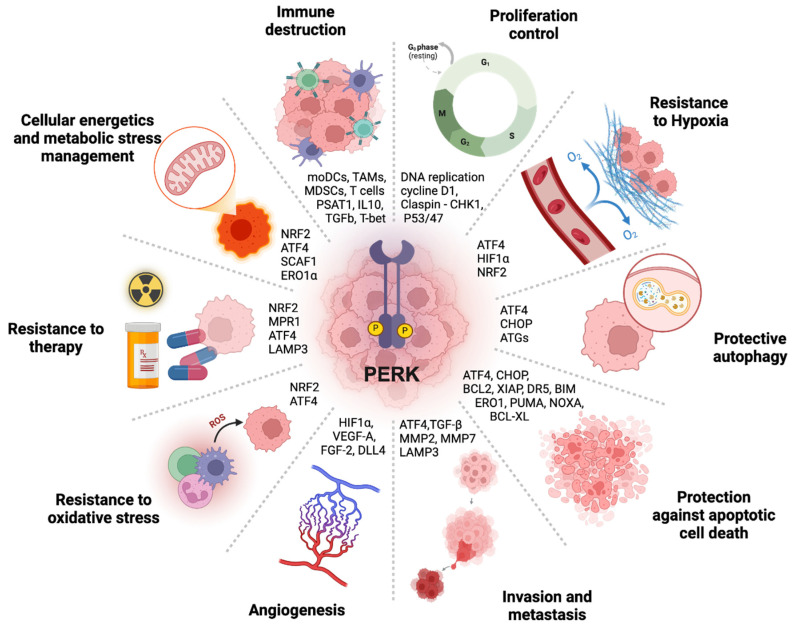
PERK pathway and the hallmarks of cancer. Tumor cells develop various characteristics that facilitate disease progression and have been defined as the “hallmarks of cancer” [95]. As pointed out in this review, PERK pathway activation plays a significant regulatory role on many of these hallmarks, including immune response, proliferation, resistance to hypoxia and oxidative stress, the promotion of cell invasion, metastasis, angiogenesis, the regulation of protective or cytotoxic autophagy, the regulation of apoptotic cell death, and resistance to various therapies. Inhibiting or, inversely, activating PERK depending on the tumoral cell context could provide new therapeutic avenues of interest in the treatment of a variety of cancers. The identified molecular or cellular mediators of PERK effects are indicated for each hallmark (see corresponding section of the main text for additional information).

**Table 1 biomolecules-15-00248-t001:** Comprehensive overview of research reports highlighting the antitumor effects induced by PERK targeting.

Cancer	PERK Function	PERK Targeting Approach	Reported Effect	References
Multiple myeloma (MM)	Cytoprotective	mRNA knockdown by siRNAs on MM cell lines cultured in vitro.	Induces autophagy-mediated cell death.	[156]
Multiple myeloma (MM)	Cytoprotective	Pharmacological inhibition with compound GSK2606414 in single treatment and combination treatment with Bortezomib (proteasome inhibitor).	Reduces MM cell viability in vitro (induction of apoptosis). Cytotoxic effect of GSK2606414 is additive to that of bortezomib in combination treatments.	[157]
Pancreatic cancer; multiple myeloma (MM)	Cytoprotective	Pharmacological inhibition with compound GSK2656157 (derivative of GSK2606414).	Inhibits the growth of human tumor xenograft in mice (decreased angiogenesis; apoptosis induction). Reported side effect of PERK targeting: pancreatic damage (reversible).	[141,158]
Pancreatic cancer	Cytoprotective	Pharmacological inhibition with compound GSK2606414.	Inhibits the growth of human tumor xenograft in mice (apoptosis induction).	[159]
Breast cancer	Stimulation of cell migration	mRNA knockdown by siRNAs on cell lines cultured in vitro.	Reduces cell migration under hypoxic conditions (transwell and gap closure assays).	[131]
Breast cancer	Cytotoxic	Pharmacological activation with compound CCT020312 in combination treatment with taxol.	Combination treatment induces taxol-mediated cell apoptosis and reduces tumor growth of human xenograft in mice.	[160]
Breast cancer	Stimulation of cell migration and invasion	Pharmacological inhibition with compound 3-Fluoro-GSK2606414.	Decreases viability in cells undergoing epithelial-to-mesenchymal transition (EMT). Reduces the ability of EMT cells to migrate and to form tumor spheres in vitro. Reduces the metastatic capacity of human tumor cells in mice (xenografts).	[124,161]
Breast cancer	Cytotoxic	Pharmacological inhibition with compound HC-5404	Impairs metastasis by killing quiescent/dormant disseminated cancer cells (DCC) in the MMTV-HER2 mouse model.	[162]
Glioblastoma	Stimulation of angiogenesis	mRNA knockdown (shRNAs)	Reduces angiogenesis in vitro (HUVEC cells, tube formation assay).	[163]
Colorectal cancer (CRC)	Cytotoxic	Pharmacological activation with compound CCT020312. (CCT) in single treatment and combination treatment with taxol (microtubule stabilizing agent, mitotic blocker)	Promotes apoptosis in vitro. CCT treatment synergistically increases the sensitivity of CRC cells to taxol, both in vitro and in vivo in mouse xenograft models.	[164]
Renal carcinoma	Cytoprotective	Pharmacological inhibition with compound 28.	Inhibits the growth of human tumor xenografts in mice.	[165]
Renal cell carcinoma (RCC)	Stimulation of angiogenesis	Pharmacological inhibition with compound HC-5404 in single treatment and combination treatment with VEGF receptor tyrosine kinase inhibitors (VEGFR-TKIs)	Single treatment reduces in vivo tumor growth. Combination treatment enhances the antiangiogenic effects of VEGFR-TKIs (human xenografts in mice) by disrupting the adaptive stress response.	[142]
Melanoma	Protects cancer cells from paraptosis-induced immunogenicity cell death (ICD)	PERK gene ablation with CRISPR-Cas9 in melanoma cell lines. Pharmacological inhibition with compound AMG44	PERK gene knockout in melanoma cells and PERK pharmacological inhibition trigger paraptosis and induction of ICD in human xenograft models in mice.	[146]
Hepatocellular carcinoma (HCC)	Cytoprotective	Pharmacological inhibition with compound GSK2656157.	Increases proteotoxic stress and reduces cancer cell viability and proliferation in vitro; decreases tumor burden in orthotopic models of HCC in mice.	[166]

## Data Availability

Not applicable.

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
