# Peer review of "PERK-Olating Through Cancer: A Brew of Cellular Decisions"

_biomolecules, 2025, doi:10.3390/biom15020248_

Round 1

Reviewer 1 Report

Comments and Suggestions for Authors

Specific comments to the authors

The submitted review "PERK-olating Through Cancer: A Brew of Cellular Decisions" provides a comprehensive overview of the multiple and heterogeneous intracellular roles and functions of PERK in relation to cancer cells and the associated microenvironment, based on previously published reviews as well as in vitro/in vivo experiments and also first clinical trials.

The topics presented range from classical cancer in relation to the endoplasmic reticulum (ER) and the three independent ER membrane-associated sensors, through cellular responses and signalling pathways activated by PERK, to the potential role of PERK in human cancer for therapeutic applications. In summary, the author provides an interesting overview of the diverse cellular functions of PERK and its potential role in human cancer carcinogenesis, as well as the impact and potentially great opportunity of PERK-based targeted therapy in human cancer now and in the future, which is mostly easy to read, follow and understand. Authors should clarify some issues before accepting the manuscript for publication, as mentioned below.

Major concerns:

Please avoid superfluous sentences and try to make short and clear statements within the manuscript.

Minor concerns:

# Title: The title should emphasise that most of the findings to date are based only on in vitro or in vivo data.

# Abstract: Please specify the human entities that definitively show deregulation of PERK.

# Figure 4: Please add the molecular "players/mediators" to show how PERK modulates the hallmarks of cancer.

# Chapter “3. A-PERK and cell autonomous modulation of cancer cell survival”: please add appropriate references to the paragraph to Chapter 3.1.

 # Chapter "4. -PERK as a target for cancer therapy: current status and possible future directions": The results should be transferred to an additional table summarising which human malignancies show deregulation (pro- or anti-carcinogenesis) of PERK (including cell line, type of experimental design (in vitro and in vivo) and drugs used).

# "5. Conclusion: This chapter is largely speculative and should include some sort of milestones. What are the limitations of known PERK-based drugs? How might the in vitro and in vivo findings of PERK in human carcinogenesis be translated into clinical application?

Comments on the Quality of English Language

The English could be improved to more clearly express the research.

Author Response

Comments 1: Please avoid superfluous sentences and try to make short and clear statements within the manuscript. 

Answer 1: We have done our best to answer your request and carefully reviewed the entire manuscript with this objective in mind. We have made a concerted effort to streamline our writing, focusing on presenting our arguments and results as clearly and concisely as possible. We believe these changes have significantly improved the readability and accessibility of our work, making it more engaging for the readers of Biomolecules.

Comments 2: The title should emphasise that most of the findings to date are based only on in vitro or in vivo data.

Answer 2: We agree that it is very important to make this clear for the reader since no PERK therapy is still applied in the clinics and all presently available data on PERK potential anti cancerous functions rely exclusively from animal models and cancer cell cultures. We do not think however that this information would be best stated in the title. This information has therefore been clearly given at the end of the abstract.

Comments 3: Abstract -  Please specify the human entities that definitively show deregulation of PERK.

Answer 3: It seems presently clear that endoplasmic reticulum (ER) stress is indeed induced at various stages in most, if not all, types of solid as well as liquid tumors. ER stress and associated UPR have emerged as a key elements regulating cancer development and progression. The targeted therapy of these pathways is likely to be applicable to a wide range of tumor types. To address the ambiguity in our initial sentence which introduced this point, the text of the abstract has been reformulated to clarify our meaning. Whatever, it is indeed highly probable that some types of cancers may be more sensitive to PERK targeting and we have previously discussed this point in the conclusion section of the manuscript

Comments 4: Figure 4 -  Please add the molecular "players/mediators" to show how PERK modulates the hallmarks of cancer.

Answer 4: For each cancer hallmark, the players (downstream PERK effectors, cellular processes) involved in the cellular response can vary significantly. This variability reflects the complex nature of the role of PERK in cancer biology. We agree with reviewer 1 that it may be very valuable for the reader to be able to appreciate this by looking directly at Figure 4. As requested, we have therefore indicated in the figure (where the information is known) the main effectors or processes involved for each hallmark. We have explained some abbreviations in the figure legend, and also invited the reader to refer to the main text of the article for more details (all molecular players and mediators cited are of course discussed extensively throughout the review).

Comments 5: Chapter “3. A-PERK and cell autonomous modulation of cancer cell survival”: please add appropriate references to the paragraph to Chapter 3.1.

Answer 5: We understand Reviewer 1's concern about the references in the section of Chapter 3.1. We would like to clarify that this paragraph is intended as a brief introduction to the chapter, providing a general overview of the topic. The subsequent sub-chapters contain all the appropriate and detailed references to support our statements. However, in response to this request and to increase the usefulness of this introductory section, we have added several recent review references relevant to this introductory section. These additions provide a general overview of the role of stress in cancer cell survival and proliferation.

Comments 6: Chapter "4. -PERK as a target for cancer therapy: current status and possible future directions": The results should be transferred to an additional table summarising which human malignancies show deregulation (pro- or anti-carcinogenesis) of PERK (including cell line, type of experimental design (in vitro and in vivo) and drugs used).

Answer 6: We have now added a new table (Table 1) giving the references of the most interesting research works where it has been shown that PERK targeting can have an antitumor effect. This table summarizes the type of human malignancy studied, experimental process used for PERK targeting, drugs used, and observed effect on cancer cells (both in vitro and in vivo). This additional table enhances the accessibility of the information, allowing readers to quickly grasp the current status of PERK as a target for cancer therapy across various cancer types.

Comments 7: 5. Conclusion: This chapter is largely speculative and should include some sort of milestones. What are the limitations of known PERK-based drugs? How might the in vitro and in vivo findings of PERK in human carcinogenesis be translated into clinical application?

Answer 7: The conclusion chapter has undergone strong reformulation and shortening since (we agree with the reviewer) the majority of the points discussed were speculative and finally not so informative after having read the two preceding chapters presented (4-1 and 4-2) which discussed in details the strategies and limitations of PERK pharmacological targeting. We therefore tried instead to provide in this very last section a concise summary of the key points relevant to the consideration of PERK targeting strategies for cancer treatment.

Reviewer 2 Report

Comments and Suggestions for Authors

In this manuscript, L. Mazzolini and C. Touriol summarize and evaluate the intracellular functions of the type I protein kinase (PERK) and their impact in growth and microenvironment of cancer cells. Further, the authors, in particular, highlight the potential therapeutic strategies that target this unique protein kinase. Within the unfolded protein response (UPR) signaling pathway, activation of PERK as one of three branches plays a crucial role in maintaining cellular homeostasis, having a direct impact on cell viability and survival. More importantly, the authors’ evaluation and comments were focused on the dual role of PERK, i.e. adaptive and cytotoxic under prolonged ER stress. The PERK’s two-faceted role suggests that OERK-based cancer therapy requires carefully consideration on the chosen targeting approaches, i.e. inhibition or activation, for the cancer targeted. This review provided some interesting insights into the research and development of PERK-targeting anticancer drugs. I would like to recommend publication of this manuscript subjected to a minor revision.

Specific comments:

1.     In my opinion, the numbered scheme for sections and subsections are in a bit mess. For instance, For sections 1, 2 and 3, the authors number subsections as “1.A-”, “1.B-” …; for subsection 3.A-, the authors number sub-subsections as “3.A1-“, “3.A2-“;  However, for section 4, the authors number subsections as “4.1.-“, “4.2.-“. I suggest that authors use uniform numbering, e.g.

1.

1.1

1.1.1

… ….

2.

2.1

2.1.2

… …

2.     Please authors check and correct typos throughout the manuscript.

A few examples:

-       Line 33: “or” should be “of”

-       Lines 41 and 42: “cell” should be “cells”

-       Line 93: add a “.” After “(5, 13)”

-       Line 872: “targeted” should be “targeting”

Author Response

Comments 1: 

  • In my opinion, the numbered scheme for sections and subsections are in a bit mess. For instance, For sections 1, 2 and 3, the authors number subsections as “1.A-”, “1.B-” …; for subsection 3.A-, the authors number sub-subsections as “3.A1-“, “3.A2-“; However, for section 4, the authors number subsections as “4.1.-“, “4.2.-“. I suggest that authors use uniform numbering, e.g.

1.

1.1

1.1.1

Answer 1: We sincerely apologize for the clumsy numbering of the chapters in our manuscript that you pointed out. We greatly appreciate your attention to detail and the time you've taken to provide this valuable feedback. Following your advice, we have carefully checked and corrected all chapter and section numbers to ensure they are accurate and logically sequenced throughout the manuscript.

Comments 2: Please authors check and correct typos throughout the manuscript.

A few examples:

  • Line 33: “or” should be “of”
  • Lines 41 and 42: “cell” should be “cells”
  • Line 93: add a “.” After “(5, 13)”
  • Line 872: “targeted” should be “targeting”

Answer 2: We also apologize for the typographical errors still present in the submitted manuscript. We understand that such errors can detract from the overall quality of the work and potentially lead to misunderstandings. Thus, in light of your feedback, we have thoroughly reviewed the entire manuscript and made (all, we expect) the necessary corrections.

Reviewer 3 Report

Comments and Suggestions for Authors

PERK-olating Through Cancer: A Brew of Cellular Decisions

Topic is very interesting. The type I protein kinase PERK is an endoplasmic reticulum (ER) transmembrane protein 9 that plays a multifaceted role in cancer development and progression, influencing tumor growth,  metastasis and cellular stress responsesPERK targeting could impair tumoral growth by a wide variety of mechanisms, involving the tumor cell itself but also its microenvironment. These include: increased tumor cell death through apoptosis or autophagy, decreased metastasis, impaired angiogenesis, reduced resistance against hypoxia, reduced tumor dormancy, enhanced immunogenic cell death and restored anti-tumor immune response. In addition, PERK targeting can also sensi- 863 tize tumoral cells to the action of various anticancer drugs. The article is systematized and the 4 figures are well done. References must be renewed.

Author Response

Comments 1: References must be renewed.

Answer 1: We sincerely thank the reviewer for his analysis of the manuscript and thank him for his positive feedback on our review article. According to his request, we have verified all references cited and have tried to include the most recent publications in our article. Indeed, we fully agree with the reviewer that the timeliness of the information provided is a key aspect to consider when writing a journal article. We have therefore removed 22 citations that were a little old and, in most cases, replaced them with more recent references. However, we retained some citations for (sometimes old) research work that has contributed to a major advance in the field. In these cases, we have ensured that the reader also has access to additional citations that provide direct information on the most important recent developments in the treated subject (key reviews or articles). 22 new references have consequently been added to the revised version. We believe that our manuscript is now fully up to date and hope that it will meet the expectations of the Biomolecules reviewer and editorial board. (For rapid evaluation: overview of the time-distribution of the 194 references cited in this review:  anterior to 2015: 73; from 2015 to present: 120  -  2015-2017 : 25/ 2018-2020: 44 /2021-2022: 23/ 2023-present: 29).

Round 2

Reviewer 2 Report

Comments and Suggestions for Authors

I appreciate that you have made effort to address my concerns  arising from your original manuscript, and would like to recommend the publication of your revised manuscript in its current form. 

Reviewer 3 Report

Comments and Suggestions for Authors

I agree with this version of the manuscript